# Unsupervised and Generic Short-Term Anticipation of Human Body Motions

**DOI:** 10.3390/s20040976

**Published:** 2020-02-12

**Authors:** Kristina Enes, Hassan Errami, Moritz Wolter, Tim Krake, Bernhard Eberhardt, Andreas Weber, Jörg Zimmermann

**Affiliations:** 1Visual Computing Department, University of Bonn Germany, 53115 Bonn, Germany; kristinaenes@hotmail.de (K.E.); errami@cs.uni-bonn.de (H.E.); wolter@cs.uni-bonn.de (M.W.); weber@cs.uni-bonn.de (A.W.); 2HdM Stuttgart, 70569 Stuttgart, Germany; tim.krake@visus.uni-stuttgart.de (T.K.); eberhardt@hdm-stuttgart.de (B.E.); 3Fachbereich Informatik, University of Stuttgart, 70569 Stuttgart, Germany

**Keywords:** dynamic mode decomposition, human motion anticipation, short-time future prediction, delay coordinates

## Abstract

Various neural network based methods are capable of anticipating human body motions from data for a short period of time. What these methods lack are the interpretability and explainability of the network and its results. We propose to use Dynamic Mode Decomposition with delays to represent and anticipate human body motions. Exploring the influence of the number of delays on the reconstruction and prediction of various motion classes, we show that the anticipation errors in our results are comparable to or even better for very short anticipation times (<0.4 s) than a recurrent neural network based method. We perceive our method as a first step towards the interpretability of the results by representing human body motions as linear combinations of previous states and delays. In addition, compared to the neural network based methods large training times are not needed. Actually, our methods do not even regress to any other motions than the one to be anticipated and hence it is of a generic nature.

## 1. Introduction

Various kinds of neural network architectures are the main technical basis for the current state of the art for anticipation of human body motions from data [1,2,3,4,5,6,7,8,9]. However, as is the case in many other application domains, there is a fundamental lack of interpretability of the neural networks. In these approaches the two main conceptual ingredients of human motion prediction are also intermixed:(a)Modelling the intent of the persons.(b)Modelling the influence of the previous motion.

Whereas for point (a), mechanistic models might be hard to obtain for point (b) models as dynamical systems partially reflecting bio-physical knowledge are possible in principle. In this paper, we will focus on point (b). Instead of suggesting another neural network based anticipation architecture, we will try to separate several possible constituents:Can recently developed so called equation free modelling techniques [10,11,12,13,14,15] already explain and predict motions in a short time horizon?What is the role of incorporating delay inputs? Many neural network architectures incorporate delays [16,17,18], recurrent connections [2,19] or temporal convolutions [5,20,21], but the contribution of the delays or the memory cannot be separated from the overall network architecture.

We show that a direct application of the equation free modelling technique of Dynamic Mode Decomposition (DMD) does not yield good results for motion prediction in general. However, when incorporating delays, the corresponding technique of Dynamic Mode Decomposition with delays (DMDd) does not only yield almost perfect reconstructions of human motions, but it is also very suitable for short-term motion anticipation! Regardless of potential applications of the method, which we will discuss in the conclusion, we show the relevance of incorporating the information given by delays.

The paper is structured as follow: In Section 2 we clarify the theoretical background of our work: Dynamic Mode Decomposition (DMD), Taken’s Theorem, and Dynamic Mode Decomposion with Delays (DMDd). In Section 3, we explain our experiments with DMDd and give some examples of our results. In Section 4, we discuss future possibilities given by our method.

## 2. Theoretical Background

First, we give a motivation for Dynamic Mode Decomposition.

In general, dynamical systems are usually described via a set of differential equations. For many systems, a variety of appropriate data in form of observables are available. However, if the process is complex the recovery of the underlying differential equation from data is a challenging task [22]. Instead, the set of *m* observables sampled at time steps *n* is used for the investigation of the considered process. For the identification of temporal structures, the Fourier theory is usually utilized. Therefore, a Fourier analysis on the observables is performed, to extract amplitude and frequencies leading to a decomposition into trigonometric series. This approach has some drawbacks for human motion capture data as these phenomena do not exclusively consist of periodic components. Hence, the decomposition will be distorted. An algorithm that take this point into account is Dynamic Mode Decomposition (DMD).

For the application on motion capture, data we assume a vector-valued time series $x_{1},x_{2},\ldots ,x_{n}\in R^{m}$, where each snapshot consists of marker positions (in 3D) or joint angles of a skeleton to a certain time step. Before we formulate the algorithm in more detail, we briefly highlight the basic concept of DMD: In a first step, the data were used to determine frequencies, the so-called DMD eigenvalues. These are defined by the non-zero eigenvalues of a solution to the following minimization problem:(1)$$\min_{A\in C^{m\times m}}\sum_{j=1}^{n}{∥Ax_{j}-x_{j+1}∥}_{2}^{2}.$$

Then, the data were fitted to the previously computed frequencies (this process is similar to a discrete Fourier transformation or a trigonometric interpolation).

However, in many application areas, the number of observables is considerably larger than the number of snapshots, i.e., m>n. Therefore, this approach leads to a sufficient number of frequencies and it can be proven that the reconstruction is error-free [23]. For motion capture data, however, the converse is true, i.e., m<n. Hence, in most cases we do not have enough frequencies for an adequate reconstruction, which results in a bad anticipation as well.

We approach this issue by manipulating the data in a preprocessing step, i.e., before applying EXDMD. To this end, the theory of delays justified by Takens’ Theorem is consulted, which is described in Section 2.2. Applying this technique leads to Dynamic Mode Decomposition with delay (DMDd) [10]. The exact procedure is explained in Section 2.3.

### 2.1. Exact Dynamic Mode Decomposition

EXDMD is the most modern variant of DMD that is applied directly on raw data. It was published in 2014 by Tu et al. [12]. However, we have chosen the algorithmic formulation by Krake et al. [23], which differs in the computation of DMD amplitudes. Algorithm 1 shows an adjusted version of the algorithm. Since we mainly focus on anticipation, we are not interested in the reconstruction of the first snapshot and therefore some steps are skipped.

After defining the snapshot matrices *X* and *Y*, which are related by one time-shift, a reduced singular value decomposition of *X* is performed in line 2. These components are used to determine the low-dimensional matrix *S* that owns the dynamic relevant information in form of (DMD) eigenvalues $\lambda_{j}$. Therefore, only the non-zero eigenvalues are used to compute the so-called DMD modes $\vartheta_{j}$ in line 7. Finally, the DMD amplitudes are calculated via $a=\Lambda^{-1}\Theta^{+}x_{2}$, where the second initial snapshot $x_{2}$ is used.

Given the DMD modes, DMD eigenvalues, and DMD amplitudes, we can both reconstruct the original snapshot matrix and make predictions for future states. But as mentioned before a good reconstruction might not be possible depending on the matrix dimensions. However if all conditions are met we can achieve an exact reconstruction.
**Algorithm 1** Exact Dynamic Mode Decomposition1:Define $X=[x_{1}\;\ldots \;x_{n-1}]$, $Y=[x_{2}\;\ldots \;x_{n}]$2:Calculate the reduced SVD $X=U\Sigma V^{*}$3:Calculate $S=U^{*}YV\Sigma^{-1}$ with rank(X)=r4:Calculate $\lambda_{1},\ldots ,\lambda_{r}$ and $v_{1},\ldots ,v_{r}$ of *S*5:**for**1≤i≤r**do**6:    **if**
$\lambda_{i}\neq 0$
**then**7:        $\vartheta_{i}=\frac{1}{\lambda_{i}}YV\Sigma^{-1}v_{i}$8:$\Lambda =\mathrm{diag}(\lambda_{1},\lambda_{2},\ldots ,\lambda_{r_{0}})$ with $\lambda_{1},\lambda_{2},\ldots ,\lambda_{r_{0}}\neq 0$9:$\Theta =[\vartheta_{1}\;\vartheta_{2}\;\ldots \;\vartheta_{r_{0}}]$10:Calculate $a=\Lambda^{-1}\Theta^{+}x_{2}$ with $a=(a_{1},\ldots ,a_{r_{0}})$

### 2.2. Delay Vectors and Takens’ Theorem

Most real world dynamical systems are only partially observable, i.e., we can observe only a low-dimensional projection of a dynamical system acting on a high dimensional state space. This means that from a certain observed snapshot of a dynamical system it is even in principle not possible to reconstruct the full current state of the dynamical system. Fortunately, the information contained in observations made at several different time steps can be combined to reconstruct, at least in principle, the complete current state, and (under certain technical assumptions) the dynamics on these delay vectors is diffeomorphic to the true dynamics on the hidden state space. This delay embedding theorem is also known as Takens’ theorem, first proved by Floris Takens in 1981 [24]. This result has led to a branch of dynamical systems theory now referred to as “embedology” [25].

Here, we give a brief sketch of the delay embedding theorem for discrete-time dynamical systems. Let the state space of the dynamical system be a *k*-dimensional manifold *M*. The dynamics is defined by a smooth map
(2)ϕ:M→M,
and the observations are generated by a twice-differentiable map y:M→R (the observation function), projecting the full state of the dynamical system to a scalar observable. From a time series of observed values, we can build *m*-dimensional *delay vectors*:(3)$$y_{m}\left(n\right)={\left(y\left(n\right),y\left(n-1\right),y\left(n-m+1\right)\right)}^{T}.$$

The delay vectors are elements of $R^{m}$ and by mapping a delay vector to its successor we get a mapping ρ from $R^{m}$ to $R^{m}$:(4)$$\rho \left(y_{m}\left(n\right)\right)=y_{m}\left(n+1\right)$$

The delay embedding theorem now implies that the evolution of points $y_{m}\left(n\right)$ in the reconstruction space $R^{m}$ driven by ρ follows (i.e., is diffeomorphic to) the unknown dynamics in the original state space *M* driven by ϕ when m=2k+1. Here 2k+1 is a maximal value, faithful reconstruction could already occur for delay vectors of lower dimension. Thus long enough delay vectors represent the full hidden state of the observed dynamical system, meaning that the prediction of the next observed value based on a long enough history of past observations becomes possible.

For our purposes, we take the delay embedding theorem as an indication that adding delay dimensions to the observed state vector can improve the anticipation quality of a DMD model.

### 2.3. Dynamic Mode Decomposition with Delays (DMDd)

Our motion capture data has the following form:(5)$$X=\left[\begin{array}{ccc} x_{1} & \ldots & x_{n} \end{array}\right]$$

Each state $x_{i}$ at time step 1⩽i⩽n, is a vector of length *m*. To augment this matrix with *d* delays we use a window of size m×(n−d), with 1<n−d<n, to move along the motion data. This window starts at the first frame of *X* and makes a copy of the first n−d frames of the data, before taking a step of one frame along *X*. We continue with this process until the window reaches the end of the motion data. The copied data are then stacked one above the other resulting in a matrix $\tilde{X}$ with n−d columns and (d+1)m rows:(6)$$\tilde{X}=\left[\begin{array}{ccc} x_{1} & \ldots & x_{n-d}\\ x_{2} & \ldots & x_{n-d+1}\\ \vdots & \ddots & \vdots \\ x_{d+1} & \ldots & x_{n} \end{array}\right]$$

Depending on how we choose *d*, the problem where our data has more columns than rows is no longer given. Applying the DMD algorithm to $\tilde{X}$ provides us with a good representation of the data and a good short-term future prediction is also possible, as will be detailed in Section 3.

## 3. Results

We tested DMDd on the Human3.6M dataset [26], which consists of different kinds of actions like walking,sitting and eating. These actions are performed by different actors. For our experiments we first choose the motion sequences performed by actor number 5 (to have comparable results to the literature, as this actor was used for testing in the neural network based approaches, whereas the motions of the other actors were used for training). The data we use is sampled at 50 Hz and contains absolute coordinates for each joint. For each experiment we divide each action sequence into several sub-sequences of 100 or 150 frames length. The first 50 (1 s) or 100 frames (2 s) are taken as input for our methods and we compute a prediction for the next 5 frames (0.1 s), 10 frames (0.2 s) and 20 frames (0.4 s). To measure the distance between the ground truth GT and our prediction *P* we use two different distance measures. The first measure we use is the mean squared error (MSE):(7)$$L\left(\mathrm{GT},P\right)=\frac{1}{K}\sum_{k=1}^{K}\frac{1}{mp}\sum_{i,j}{\left({\mathrm{GT}}_{ijk}-P_{ijk}\right)}^{2}$$
*K* is the number of motion sequences taken form the same action class and hence the number of predictions made for this action. Both GT and *P* consist of *m* observables and *p* frames. The second distance measure we use is the Kullback-Leibler divergence as it was used in [8].

### 3.1. Comparison with Neural Network Based Methods

First we compare the setting of having the information of 1 s of motions as inputs (50 frames) using DMD with 80 delays with a RNN baseline as the one used in [27]. We use the mean squared error (MSE) as well as the Kullback-Leibler divergence as error measures for anticipation times of 0.1 s, 0.2 s, and 0.4 s.

The results given in Table 1 and Table 2 indicate that our method shows better results for 0.1 s, 0.2 s and for most motion classes even for 0.4 s, although they are not only unsupervised but even no knowledge about any other motion is taken into account! Interestingly, the error of the RNN slightly decreases with the anticipation times. This counter-intuitive behavior of the RNN approach might be explained by the fact that the anticipations yielded by the RNN baseline in general shows small jumps at the beginning of the anticipation period [28].

### 3.2. Reconstruction and Anticipation of Motions Using DMD and DMD with Delays

Adding time delays already improves the reconstructibility of motions. In Table 3, we show the average reconstruction errors of motion clips of 2 s length (100 frames) for the different motion classes. Already adding 10 time delays yields a dramatic improvement. After adding 60 delays the reconstruction error drops to less than $10^{-5}$ for all motion classes.

The results of the anticipation errors for 0.4 s (20 frames) of anticipation using 2 s (100 frames) as context length is given in Table 4. The anticipation errors for DMD without delays is large (>$10^{10}$ for all motion classes and is not reproduced in the table. In contrast to the reconstruction case, in which the error monotonically decreases with adding additional delays, the anticipation errors have minima at a certain number of delays (ranging between 40 and 90 for the different motion classes.

In Figure 1, we give skeleton visualizations of a walking motion, and the anticipated skeleton frame for 2 s of input motion length and 0.4 s of anticipation time. Visually the anticipated skeleton pose is not distinguishable from the ground truth skeleton pose.

### 3.3. Using Different Input Lengths of Motions to Be Anticipated

We compare the previously used setting of having the information of 1 s of motions as inputs (50 frames) to the one with 2 s of motions as inputs (100 frames), and 4 s of motions as inputs (200 frames). In Figure 2, we show the MSE for the anticipation of a trained RNN with 1 s of motions as inputs, the DMDd with 1 s of motions as inputs, DMDd with 2 s of motions as inputs, and DMDd with 4 s of motions as input (for anticipation times of 0.1 s, 0.2 s, and 0.40 s).

### 3.4. Reconstruction and Anticipation of Inertial Measurements of Root and End-Effectors

For assessing short term anticipation on the basis of sparse acceplerometer data attached to the end effectors and the hip we used the marker position data of the Human 3.6M database to have a large collection of motions and “ground truth data”. As it has already been shown in [29] using the second time derivatives of marker position data yields reliable estimates for tests using data of accelerometers.

In Figure 3, the results of the anticipation error of just the marker of the right hand is given. The anticipation error of performing the DMDd80 on the time series of just this one marker is given as M1F1. The error of this one marker but using DMDd80 on the end effectors and the root is given as M5F1; the one performing DMDd80 on all 17 markers is given as M17F1. Using second derivatives as simulation of accelerometer sensor data are given similarly as A1F1, A5F1, and A17F1. Whereas the addition of a “spatial context” of other markers than the one measured for anticipation in the DMDd computation has little effect for 0.1 s of anticipation time, there is a considerable effect for 0.2 s of anticipation time, and a huge effect for 0.4 s of anticipation time: For the simulated accelorometers the corresponding results are given in Figure 4. The average MSE of the right hand marker’s accelerations with a value of 553,000 was about four orders of magnitude larger when performing the DMDd only on its time series compared to the one using the spatial context of the 4 additional ones (left hand, left and right foot, and root) with a value of 14. Considering more than 4 additional markers had little additional effect.

## 4. Conclusions and Future Work

In contrast to some special classes of human motions, on which the direct application of DMD to the observables of human motion data can be suitable for a good reconstruction of the data [13,30], these direct applications of DMD to the observables of the motions contained in the Human 3.6M dataset do not yield good reconstructions, nor suitable short-term anticipations.

Inspired by Takens’ theorem, which emphasizes the usefulness of delays in reconstructing partially observable, high dimensional dynamics, we have extended DMD with delay vectors of different length and evaluated the impact on short-term anticipation using a large real world human motion data base and comparing the performance to a state of the art RNN model. The results show that delays can drastically improve reconstruction and also anticipation performance, often by several orders of magnitude, and, in many cases, lead to better anticipation performance than the RNN model (for anticipation times less than 0.4 s). This is especially remarkable, as our methods do not even regress to any other motions than the one to be anticipated. Moreover, DMD effectively solves a convex optimization problem and thus is much faster to evaluate than training RNNs. Additionally, solutions of convex optimization problems are globally optimal, a guarantee which is absent for trained RNNs.

As already mentioned in the introduction, the presented work was primarily concerned with modelling the influence of the previous motion on motion anticipation. For modelling the *intent of persons*, other methods are required, and neural network based methods might be the ones of choice. Coming up with a hybrid DMD and neural network based method for mid-term (or even long-term) motion anticipation will be the topic of future research.

Direct applications of our work are feasible. As our methods are generic and require much less computational resources than neural network based techniques, they are well suited to be used with mobile robots and their limited computation power. A short term anticipation of human body poses might be used for safety checks not to harm any body part of non-static humans when operating close to them.

Additionally in robotic applications we require safety guarantees. Such assurances are very hard to give and hard to prove for large non-linear, non-convex-machine learning models. The convex and in essence linear DMD methods have been neatly integrated into modern control theory [10]. We therefore advocate the use of DMD in short term scenarios where safety guarantees are paramount.

As a final remark, we mention that linear methods like DMD can foster the interpretability of results by representing the evolution of motion as a linear combination of “factors”, where factors can be previous states, delays, or nonlinear features computed from the previous states or delays. This could prove to be especially useful when machine learning driven systems enter more and more critical application areas, involving aspects of security, safety, privacy, ethics, and politics. To address these concerns, and for many application areas involving anticipation of human motions these concerns play a central role, transparency, explainability, and interpretability become more and more important criteria for the certification of machine learning driven systems. For a comprehensive review of the current literature addressing these rising concerns about safety and trustworthiness in machine learning see [31].

## Figures and Tables

**Figure 1 sensors-20-00976-f001:**
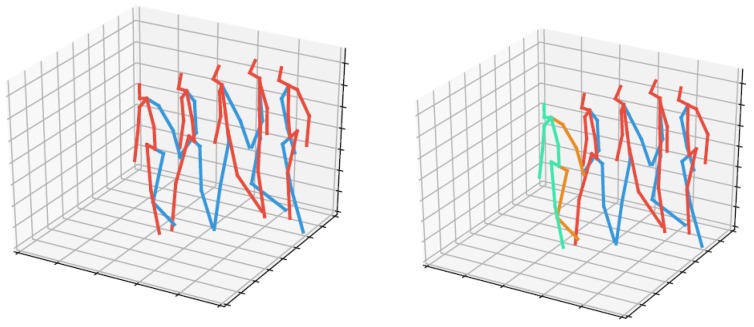
Skeleton visualization of a walking motion, and the anticipated frame of the full skeletonskeleton frame for 2 s of input motion length and 0.4 s of anticipation time. (**Left**) ground truth motion, i.e. the red-blue frames are recorded. (**Right**) initial motion segment and anticipated motion, the red-blue frames are from the recorded initial segment, the green-orange frame is the skeleton frame of the 17 markers anticipated by DMDd80.

**Figure 2 sensors-20-00976-f002:**
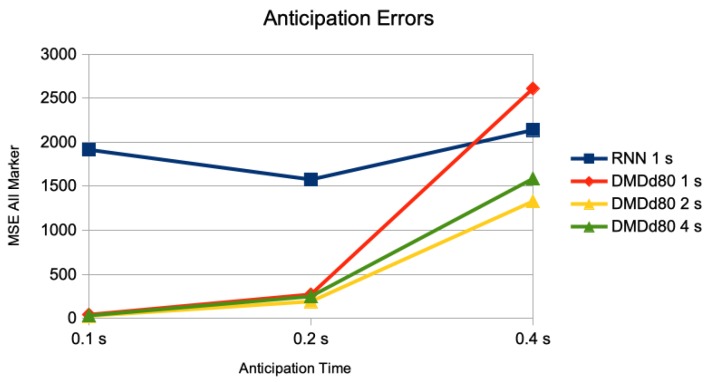
Comparison of anticipation errors for anticipations of 0.1 s (5 frames), 0.2 s (10 frames), and 0.4 s (20 frames). The result of a trained RNN using inputs of 1 s, and DMDd80 on inputs of 1 s, 2 s, and 4 s. The error measure is the average of the mean squared error on the pose sequences on the different motion classes of the Human 3.6M dataset for actor #5.

**Figure 3 sensors-20-00976-f003:**
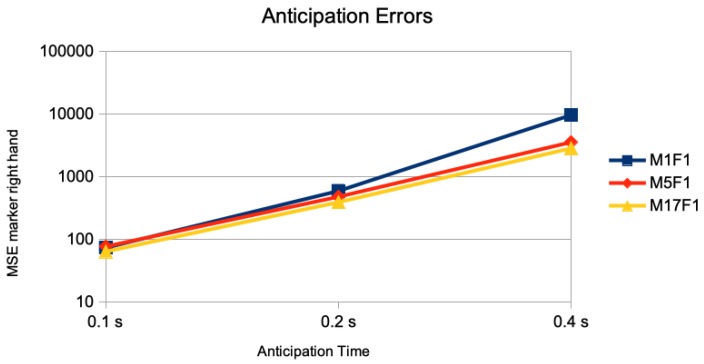
Single marker anticipation without and with spatial context. The anticipation errors of 0.1 s (5 frames), 0.2 s (10 frames), and 0.4 s (20 frames) are given for the anticipation of marker of the right hand joint using no spatial context (M1F1), all end effector markers and the root position as spatial context (M5F1), and all 17 markers as spatial context (M17F1). The error measure is the average of the mean squared error on the marker sequences of the right hand joint on the different motion classes of the Human 3.6M dataset for actor #5.

**Figure 4 sensors-20-00976-f004:**
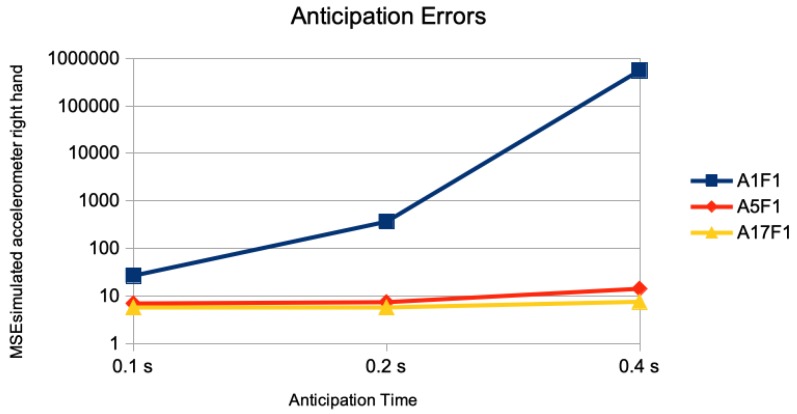
Single marker acceleration anticipation without and with spatial context. The anticipation errors of 0.1 s (5 frames), 0.2 s (10 frames), and 0.4 s (20 frames) are given for the anticipation of simulated accelerations of the marker of the right hand joint using no spatial context (M1F1), all end effector markers and the root position as spatial context (M5F1), and all 17 markers as spatial context (M17F1). The error measure is the average of the mean squared error on the accelerations marker sequences of the right hand joint on the different motion classes of the Human 3.6M dataset for actor #5. Notice that the error is given in logarithmic scale.

**Table 1 sensors-20-00976-t001:** Comparison of anticipation error using a RNN and Dynamic Mode Decomposition with 80 delays for various anticipation times. The error measure is the mean squared error of 17 markers on the pose sequences for anticipation times of 0.1 s (5 frames), 0.2 s (10 frames), and 0.4 s (20 frames) on the different motion classes of the Human 3.6M dataset for actor #5. The squared errors are expressed in ${\mathrm{mm}}^{2}$, but notice that the error is measured in $R^{3\cdot 17}$.

Action	RNN 0.1 s	RNN 0.2 s	RNN 0.4 s	DMDd 0.1 s	DMDd 0.2 s	DMDd 0.4 s
Directions	787.18	687.42	1202.92	9.09	103.03	1097.34
Discussion	964.55	874.03	1536.45	26.1	187.37	1459.77
Eating	626.26	490.74	599.55	13.16	119.89	1250.54
Greeting	1316.82	1337.01	2748.43	46.86	471.66	4096.82
Phoning	832.94	699.94	945.87	15.6	150.46	6755.74
Posing	1184.45	1118.47	2067.22	16.12	193.23	1981.44
Purchases	1176.26	1069.58	1905.41	74.84	510.89	5717.04
Sitting	1008.65	917.76	1390.35	14.05	86.67	665.49
SittingDown	11,164.34	8584.16	8784.89	207.44	750.49	4183.22
Smoking	778.76	610.34	754.22	14.37	95.79	789.76
Photo	2705.89	2070.79	2576.26	15.09	113.18	1038.58
Waiting	1081.52	886.59	1314.19	18.77	169.84	1862.09
Walking	821.12	635.70	776.37	53.82	397.00	3054.54
WalkDog	3083.24	2730.66	4248.07	91.59	588.56	3851.82
WalkTogether	1163.97	919.58	1198.45	15.00	144.91	1306.06

**Table 2 sensors-20-00976-t002:** Comparison of anticipation error using a RNN and Dynamic Mode Decomposition with 80 delays for various anticipation times. The error measure is the Kullback-Leibler divergence for anticipation times of 0.1 s (5 frames), 0.2 s (10 frames), and 0.4 s (20 frames) on the different motion classes of the Human 3.6M dataset for actor #5.

Action	RNN 0.1 s	RNN 0.2 s	RNN 0.4 s	DMDd 0.1 s	DMDd 0.2 s	DMDd 0.4 s
Directions	0.38	0.18	0.07	0.02	0.02	0.03
Discussion	0.40	0.13	0.07	0.01	0.02	0.03
Eating	0.37	0.15	0.06	0.03	0.04	0.07
Greeting	0.53	0.17	0.10	0.04	0.03	0.04
Phoning	0.26	0.16	0.07	0.02	0.03	0.03
Posing	0.50	0.19	0.10	0.02	0.03	0.03
Purchases	0.32	0.14	0.08	0.01	0.02	0.02
Sitting	0.57	0.28	0.10	0.02	0.02	0.04
SittingDown	0.74	0.35	0.17	0.01	0.02	0.03
Smoking	0.45	0.14	0.06	0.02	0.02	0.02
Photo	0.54	0.27	0.12	0.02	0.02	0.03
Waiting	0.28	0.12	0.05	0.01	0.03	0.03
Walking	0.27	0.12	0.06	0.03	0.03	0.03
WalkDog	0.48	0.19	0.09	0.03	0.03	0.04
WalkTogether	0.48	0.18	0.08	0.03	0.04	0.04

**Table 3 sensors-20-00976-t003:** Comparison of reconstructions errors using Dynamic Mode Decomposition (DMD) and Dynamic Mode Decomposition with delays for various numbers of delays (10, 20, 30, 40, 50, and 60). The error measure is the mean squared error on the pose sequences of 2 s length on the different motion classes of the Human 3.6M dataset for actor #5.

Action	DMD	DMDd10	DMDd20	DMDd30	DMDd40	DMDd50	DMDd60
Directions	3.20 × 10${}^{13}$	1.11 × 10${}^{1}$	5.99 × 10${}^{-3}$	2.56 × 10${}^{-5}$	2.22 × 10${}^{-5}$	1.15 × 10${}^{-5}$	7.61 × 10${}^{-6}$
Discussion	2.20 × 10${}^{40}$	5.02 × 10${}^{-4}$	7.93 × 10${}^{-5}$	3.76 × 10${}^{-5}$	3.39 × 10${}^{-5}$	1.51 × 10${}^{-5}$	9.89 × 10${}^{-6}$
Eating	1.61 × 10${}^{8}$	1.07 × 10${}^{-4}$	5.54 × 10${}^{-5}$	3.42 × 10${}^{-5}$	1.90 × 10${}^{-5}$	1.21 × 10${}^{-5}$	7.84 × 10${}^{-6}$
Greeting	2.77 × 10${}^{7}$	8.85 × 10${}^{-4}$	9.43 × 10${}^{-5}$	3.57 × 10${}^{-5}$	2.14 × 10${}^{-5}$	1.07 × 10${}^{-5}$	7.80 × 10${}^{-6}$
Phoning	1.70 × 10${}^{18}$	7.55 × 10${}^{9}$	1.74 × 10${}^{8}$	1.01 × 10${}^{3}$	3.30 × 10${}^{-3}$	1.73 × 10${}^{-5}$	7.46 × 10${}^{-6}$
Posing	1.00 × 10${}^{50}$	9.75 × 10${}^{3}$	5.03 × 10${}^{-4}$	9.42 × 10${}^{-5}$	1.71 × 10${}^{-5}$	1.42 × 10${}^{-5}$	8.38 × 10${}^{-6}$
Purchases	7.94 × 10${}^{44}$	7.30 × 10${}^{-4}$	6.63 × 10${}^{-5}$	2.52 × 10${}^{-5}$	2.58 × 10${}^{-5}$	1.70 × 10${}^{-5}$	6.78 × 10${}^{-6}$
Sitting	3.45 × 10${}^{27}$	9.67 × 10${}^{7}$	2.15 × 10${}^{8}$	1.95 × 10${}^{6}$	5.61× 10${}^{0}$	1.57 × 10${}^{-5}$	3.46 × 10${}^{-6}$
SittingDown	1.09 × 10${}^{45}$	4.64 × 10${}^{5}$	1.19 × 10${}^{-4}$	1.41 × 10${}^{-5}$	1.11 × 10${}^{-5}$	6.51 × 10${}^{-6}$	3.34 × 10${}^{-6}$
Smoking	1.32 × 10${}^{20}$	2.45 × 10${}^{0}$	1.95 × 10${}^{-2}$	1.15 × 10${}^{-4}$	2.18 × 10${}^{-5}$	1.18 × 10${}^{-5}$	6.70 × 10${}^{-6}$
Photo	7.66 × 10${}^{24}$	1.27 × 10${}^{0}$	6.56 × 10${}^{-3}$	3.47 × 10${}^{-5}$	4.04 × 10${}^{-5}$	2.16 × 10${}^{-5}$	8.31 × 10${}^{-6}$
Waiting	3.27 × 10${}^{27}$	6.59 × 10${}^{1}$	5.86 × 10${}^{-1}$	2.26 × 10${}^{-4}$	6.55 × 10${}^{-5}$	1.49 × 10${}^{-5}$	4.43 × 10${}^{-6}$
Walking	5.12 × 10${}^{13}$	1.07 × 10${}^{-4}$	7.83 × 10${}^{-5}$	3.57 × 10${}^{-5}$	3.05 × 10${}^{-5}$	2.51 × 10${}^{-5}$	1.51 × 10${}^{-5}$
WalkDog	3.27 × 10${}^{19}$	2.37 × 10${}^{5}$	1.68 × 10${}^{-4}$	5.02 × 10${}^{-5}$	2.47 × 10${}^{-5}$	2.42 × 10${}^{-5}$	1.07 × 10${}^{-5}$
WalkTogether	7.34 × 10${}^{6}$	7.36 × 10${}^{-5}$	7.03 × 10${}^{-5}$	1.87 × 10${}^{-5}$	3.30 × 10${}^{-5}$	2.28 × 10${}^{-5}$	1.18 × 10${}^{-5}$

**Table 4 sensors-20-00976-t004:** Comparison of anticipation errors for 0.4 s (20 frames) using Dynamic Mode Decomposition with delays for various numbers of delays (10, 20, 40, 50, 60, 70, 80 and 90). The error measure is the mean squared error on the pose sequences of 2 s length on the different motion classes of the Human 3.6M dataset for actor #5.

Action	DMDd10	DMDd20	DMDd40	DMDd50	DMDd60	DMDd70	DMDd80	DMDd90
Directions	1.87 × 10${}^{4}$	1.84 × 10${}^{3}$	1.03 × 10${}^{3}$	1.13 × 10${}^{3}$	9.29 × 10${}^{2}$	8.46 × 10${}^{2}$	8.39 × 10${}^{2}$	8.45 × 10${}^{2}$
Discussion	2.30 × 10${}^{3}$	1.47 × 10${}^{3}$	1.38 × 10${}^{3}$	1.31 × 10${}^{3}$	1.24 × 10${}^{3}$	1.23 × 10${}^{3}$	1.19 × 10${}^{3}$	1.16 × 10${}^{3}$
Eating	1.62 × 10${}^{3}$	8.71 × 10${}^{2}$	6.24 × 10${}^{2}$	7.48 × 10${}^{2}$	7.29 × 10${}^{2}$	7.23 × 10${}^{2}$	7.11 × 10${}^{2}$	6.93 × 10${}^{2}$
Greeting	4.56 × 10${}^{3}$	2.86 × 10${}^{3}$	2.53 × 10${}^{3}$	2.64 × 10${}^{3}$	2.51 × 10${}^{3}$	2.20 × 10${}^{3}$	1.99 × 10${}^{3}$	1.99 × 10${}^{3}$
Phoning	2.13 × 10${}^{13}$	2.30 × 10${}^{13}$	1.87 × 10${}^{3}$	1.25 × 10${}^{3}$	9.71 × 10${}^{2}$	8.59 × 10${}^{2}$	8.12 × 10${}^{2}$	8.29 × 10${}^{2}$
Posing	2.71 × 10${}^{8}$	2.14 × 10${}^{3}$	1.90 × 10${}^{3}$	1.45 × 10${}^{3}$	1.20 × 10${}^{3}$	1.06 × 10${}^{3}$	1.06 × 10${}^{3}$	1.06 × 10${}^{3}$
Purchases	2.64 × 10${}^{3}$	3.28 × 10${}^{3}$	2.79 × 10${}^{3}$	2.22 × 10${}^{3}$	2.09 × 10${}^{3}$	1.51 × 10${}^{3}$	1.35 × 10${}^{3}$	1.30 × 10${}^{3}$
Sitting	6.07 × 10${}^{12}$	5.31 × 10${}^{13}$	2.43 × 10${}^{3}$	8.01 × 10${}^{2}$	5.88 × 10${}^{2}$	5.81 × 10${}^{2}$	5.69 × 10${}^{2}$	5.64 × 10${}^{2}$
SittingDown	1.64 × 10${}^{10}$	1.85 × 10${}^{3}$	1.42 × 10${}^{3}$	1.31 × 10${}^{3}$	1.14 × 10${}^{3}$	1.11 × 10${}^{3}$	1.09 × 10${}^{3}$	1.13 × 10${}^{3}$
Smoking	3.83 × 10${}^{3}$	9.47 × 10${}^{2}$	8.08 × 10${}^{2}$	8.30 × 10${}^{2}$	8.04 × 10${}^{2}$	7.49 × 10${}^{2}$	7.65 × 10${}^{2}$	7.21 × 10${}^{2}$
Photo	1.00 × 10${}^{4}$	2.00 × 10${}^{3}$	1.50 × 10${}^{3}$	1.48 × 10${}^{3}$	1.35 × 10${}^{3}$	1.22 × 10${}^{3}$	1.16 × 10${}^{3}$	1.11 × 10${}^{3}$
Waiting	1.90 × 10${}^{6}$	1.30 × 10${}^{4}$	1.27 × 10${}^{3}$	1.18 × 10${}^{3}$	1.20 × 10${}^{3}$	1.19 × 10${}^{3}$	1.19 × 10${}^{3}$	1.18 × 10${}^{3}$
Walking	2.99 × 10${}^{3}$	1.72 × 10${}^{3}$	1.70 × 10${}^{3}$	1.87 × 10${}^{3}$	2.06 × 10${}^{3}$	2.05 × 10${}^{3}$	2.07 × 10${}^{3}$	2.12 × 10${}^{3}$
WalkDog	8.73 × 10${}^{10}$	4.14 × 10${}^{3}$	3.95 × 10${}^{3}$	3.68 × 10${}^{3}$	3.82 × 10${}^{3}$	3.80 × 10${}^{3}$	4.05 × 10${}^{3}$	4.75 × 10${}^{3}$
WalkTogether	1.35 × 10${}^{3}$	9.91 × 10${}^{2}$	9.93 × 10${}^{2}$	1.04 × 10${}^{3}$	1.09 × 10${}^{3}$	1.11 × 10${}^{3}$	1.08 × 10${}^{3}$	1.12 × 10${}^{3}$

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
