# Peer review of "Unsupervised and Generic Short-Term Anticipation of Human Body Motions"

_sensors, 2020, doi:10.3390/s20040976_

Round 1
Reviewer 1 Report
The paper describes a method that represents human motion in terms of factors, and uses these factors for short-term anticipations. It is well written and clearly explained, and the results are adequately presented. Using DMDd to consider not only instant data, but motion history, is a nice idea and it seems to work quite ok.
I agree with the authors in the importance of the use of these methods for certain scenarios: while nowadays it seems that everything is solved using neural networks, it is important to remind that they have drawbacks (that the authors highlight), and solutions like the one presented are sometimes much more appropiate. Moreover, the idea of the authors about using hybrid systems in the future is certainly worthy to research, in my opinion.
Overall, I like the paper, and my only concern, apart from some minor typos and mispells, is that it would be great to increase the results section with tests performed over different subjects in the Human3.6M datasets (not only subject #5).
Author Response
We have rewritten the introduction, added a pragraph, and corrected typos and mispells.
The results section was extended with a visualization of a walking sequence of length 2.4 sec using representative poses. We contrast the result of the ground truth visualization to one performing 0.4 sec of anticipation using 2 sec of context.
Reviewer 2 Report
This research tested dynamic mode decomposition with delays to predict short term human body motion. I have following suggestions.
The Introduction is not well written. For example, what are these "factors"? Authors mentioned the traditional neural network methods lack the explanation of the network and results. But the introduced method didn't show advantages either. Page 2 line 46, the proposed DMDd method works for short term only. This is a big limitation. Authors should discuss the long time prediction. Page 4 line 105, this sentence needs to be reworded, and typo "out". Page 5 Table 1, what is the unit? What are these errors? joint coordinates or joint angles? Page 7 Figure 2, x and y title fonts are too small to read. Author predicted motions but didn't show any motion graphs.
My major concern is that the prediction focuses only on short term (<0.4 second). What is the real application of such a short prediction, authors need to demonstrate that.
Round 2
Reviewer 2 Report
Authors have addressed my comments. Figure 1 is interesting and I have a question for that. For the last posture, why only half body are predicted (green) and the left arm and leg are still original motion segment (red)? Can the proposed method predict a whole-body posture in a short time window?
Author Response
The full body pose is predicted. We have clarified the description
accordingly. Ground truth frames are colored red and blue, the
predicted frame is colored green and orange. If the color of orange
is not sufficiently distinguishable from red we are happy to provide
a new coloring scheme.